# Research practice, satisfaction, motivation, and challenges among university academics in Kurdistan Region of Iraq

Hamdia Mirkhan Ahmed[1], Nazdar Ezzaddin Alkhateeb[2], Nazar P. Shabila [3,4]*, Amir Abdulrahman Ahmad[5]

1 College of Health Sciences, Hawler Medical University, Erbil, Kurdistan Region of Iraq, 2 Department of Medical Education, College of Medicine, Hawler Medical University, Erbil, Kurdistan Region of Iraq, 3 Department of Medical Laboratory Sciences, Catholic University in Erbil, Erbil, Kurdistan Region of Iraq, 4 Department of Community Medicine, College of Medicine, Hawler Medical University, Erbil, Kurdistan Region of Iraq, 5 Ministry of Higher Education and Scientific Research, Kurdistan Regionional Government, Erbil, Iraq

* nazarshabila@gmail.com

## Abstract

### Background

Researchers in universities and academic institutions must be in a leading position in generating research evidence to inform and direct national policies and strategies, improve service delivery, and achieve the main objectives. This study aimed to determine the factors that promote or hinder research productivity and quality among university academics in Kurdistan Region of Iraq.

### Methods

A cross-sectional study was conducted on 949 university academics from all public universities in the Kurdistan Region of Iraq. The authors developed a questionnaire that included sociodemographic data, challenges, satisfaction, and motivation for conducting research. Data were collected using a Google form. Frequencies, percentages, and the Chi-square test were used to analyze the data.

### Results

Most university academics (94.6%) believed that research was part of their job, but only 51.6% were satisfied with their role as academic researchers. The lack of financial motivation was the main reason for dissatisfaction, while the main incentive to conduct research was the passion for science. Around 21% of the university academics had not published any research, while 53.1% published 1–5 articles. Half of the participants (49.7%) lacked training in writing research proposals, and the majority (86.1%) have not applied for international grants. Approximately half of university academics (46.9%) shared their research findings with stakeholders, and the primary method was by sharing their published papers (59.4%), followed by seminars (42.2%). One of the important challenges in conducting research was the lack of funding (62.8%).

**Data Availability Statement:** All data files are available from the Mendeley database (https://10. 17632/bynrcrwm3r.1).

**Funding:** The author(s) received no specific funding for this work.

**Competing interests:** The authors have declared that no competing interests exist.

## Conclusions

The academics at universities in the Kurdistan Region of Iraq are passionate about their role as researchers, but face many challenges in conducting effective research. A strategic plan is needed to provide an encouraging environment for university academics regarding infrastructure, financial, and technical support. More studies are needed to identify the root factors of academic staff needs and challenges.

## Introduction

Research is essential to influence national policies and strategies that will help different countries achieve targets for different sectors such as health, education, and the related Sustainable Development Goals set by the United Nations. Researchers at universities and academic institutions should be in a leading position in generating research evidence to inform and direct national policies and strategies, improve the delivery of services, and achieve the main goals [1, 2]. Scientific research is also a vital element of success in the academic field [3]. Research productivity and quality in universities and academic institutions are crucial to their rate of contribution to creating new knowledge [4].

Published papers are the main output of scientific research. These articles are the main tools to convey new scientific discoveries to relevant authorities in the country and the rest of the world [5]. The number of published articles, articles published in reputed and indexed academic journals, and research funds are the primary measures of academic research performance [6].

Several factors hinder the production and quality of academic research. Factors promoting or hindering research productivity and quality can generally be classified into two main groups: individual and institutional factors. Individual factors include the researcher's gender, age, academic rank, salary, years of experience, teaching load, and confidence in writing research works. Institutional factors include the allocation of research funds, the size of the institution, the presence of research groups, institutional and departmental support, access to journals, the availability of research facilities, and the availability of information technology [7–9].

Many developing countries have limited opportunities for training, capacity building programs for academic researchers, and research dissemination opportunities such as conferences and symposia [10–12]. The number of academic and university researchers is constantly increasing in most countries. However, these researchers need to be adequately mentored and nurtured to be able to compete internationally. This will help reduce inequalities in research products between developed and developing countries. [11, 13]. This support can be delivered through university capacity development programs, mentoring, short- and long-term training courses, exchange programs, symposia, and conferences [10].

Another important challenge is the limited use of research evidence and findings to inform and direct national policy actions and programs [10, 13, 14]. Poor dissemination of research findings, non-alignment of research studies with governmental priorities, and poor uptake of research evidence by policymakers hinder translating newly created knowledge and findings into policy and practice [15].

In each country, it is very important to explore and identify factors that promote or hinder the productivity and quality of academic research. This will help guide the planning and implementation of appropriate interventions to promote research productivity and quality by

strengthening research capacity [16]. Academic researchers play a crucial role in creating new knowledge. Therefore, determining the factors that influence the productivity and quality of these academic researchers' research is required to develop innovative research and create new knowledge to help develop appropriate national policies and strategies in different sectors. An in-depth understanding of the obstacles faced by academic researchers is crucial for most developing countries, including Kurdistan Region of Iraq [17].

In Kurdistan Region of Iraq, no research has examined the challenges and factors linked to research productivity and quality among academic researchers. Most of the available research on this topic is related to developed countries, with limited research from developing countries [18]. These types of studies are critical in developing countries with limited funding for research. Limited resources and funds are usually directed to support university educational goals rather than scientific research [19]. Therefore, this study aimed to assess academic practice, satisfaction, motivation, and challenges in universities in the Kurdistan Region of Iraq. The findings might help develop plans to promote the quantity and improve the quality of research in the region.

## Methods

### Study design

A cross-sectional survey-based study was conducted in Kurdistan Region of Iraq from October 2021 to January 2022.

### Questionnaire

A questionnaire was developed and designed by the authors and composed of sociodemographic data and close-ended questions on the time spent by academic staff to conduct and publish research, the challenges they faced, satisfaction with their role as a researcher, their motivation to conduct research, applying for international grants, their view and expectation about the role of the Ministry of Higher Education and Scientific Research to support researchers. The study questionnaire is provided as a supporting information file (S1 Appendix).

The study questionnaire was pilot-tested on 10 participants to assess its clarity, comprehensibility, acceptance, and internal consistency. Its reliability was assessed using a test-retest approach. The Kappa statistic was calculated, which showed a reliability coefficient of 0.80. Five experts in the field evaluated the content and face validity.

Data were collected using a Google form and distributed through email and social media, such as the Viber and WhatsApp groups of academic staff, supported by high authorities of the Ministry of Higher Education and Scientific Research of Kurdistan Region of Iraq.

### Participants

A total of 1500 university academics from all public universities in Kurdistan Region of Iraq were invited to participate in the study. There are 14 universities in the region that have 10819 academic staff.

### Statistical analysis

Data were entered in the statistical package for the social sciences (SPSS) version 21. Frequencies and percentages were calculated and displayed. A Chi-square test was used to compare proportions. A *P-value* of <0.05 was considered statistically significant.

### Ethical approval

The Research Ethics Committee of Hawler Medical University approved the study (number 9/37 dated September 6, 2021). At the beginning of the questionnaire, the study objectives were explained to the academic staff, and written informed consent was obtained. Participation was voluntary, and the anonymity of the study information was guaranteed.

## Results

A total of 949 university academics responded to the survey (response rate 63.3%). Fig 1 shows the main characteristics of the sample.

Regarding the time university academics allocate for research, around one-third of university academics allocate 25% of their working hours for research. Around half of the participants (47.9%) read only 1–2 articles per week. Most academic staff stated that they had access to articles and references necessary to conduct research, although only 8.2% have access through institutional subscriptions (Table 1).

Almost all university academics believe that research is part of their job, but only 51.6% are satisfied with their role as academic researchers. The absence of financial motivation was the

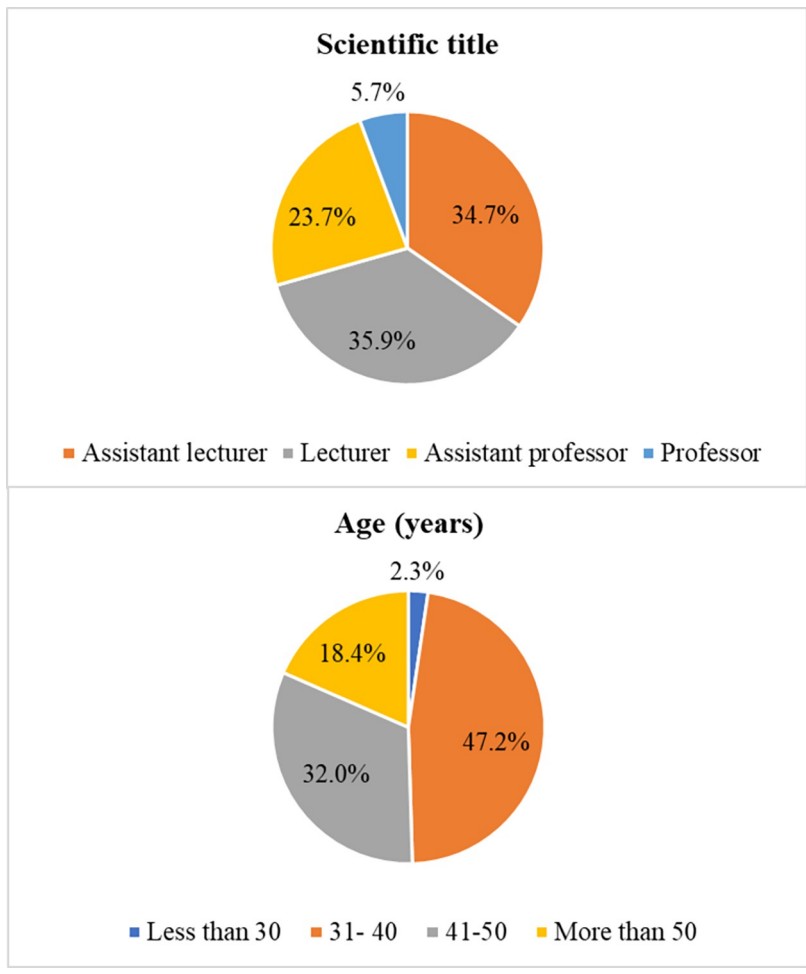

**Fig 1. Main characteristics of the study sample.**

**Table 1. Data on research and teaching activity.**

| Items | No. | (%) |
|---|---|---|
| **The proportion of work hours assigned to research** | | |
| Less than 25% | 228 | (24.0) |
| 25% | 320 | (33.7) |
| 50% | 267 | (28.1) |
| 75% | 114 | (12.0) |
| 100% | 20 | (2.1) |
| **Hours per week spent on teaching and lecturing** | | |
| 1–5 | 123 | (13.0) |
| 6–10 | 345 | (36.4) |
| 11–15 | 298 | (31.4) |
| More than 15 | 183 | (19.3) |
| **Number of articles read per week in the field of interest** | | |
| None | 52 | (5.7) |
| 1–2 | 455 | (47.9) |
| 3–4 | 231 | (24.3) |
| 5–6 | 89 | (9.4) |
| More than 6 | 120 | (12.6) |
| **Adequate access to the publications to undertake research** | | |
| No | 361 | (38.0) |
| Yes | 588 | (62.0) |
| **If yes, the nature of access** | | |
| Using free articles | 422 | (44.5) |
| Institutional/University Subscription | 78 | (8.2) |
| Personal Subscription | 165 | (17.4) |

main reason for dissatisfaction, while the main incentive for conducting research was the passion for science (Table 2).

Almost half of university academics published 1–5 articles in national or international journals, but only 5% published more than 15 articles. One-third stated that they do not have research collaboration at national or international levels, and half did not have training in writing research proposals. Most academic staff have not applied for international grants (Table 3).

The highest percentage of academic staff believe that the Ministry of Higher Education and Scientific Research has no sufficient vision, mission, or plan for conducting effective research. Regarding the attitude of academic staff towards the use of research results by stakeholders, only half of them shared their research findings with stakeholders, and the main method is through the sharing of their published articles (59.4%), followed by seminars (42.2%) (Table 4).

One of the main challenges in conducting research was the lack of funding. Table 5 shows the perceptions of university academics of the main challenges faced when conducting research in the Kurdistan Region of Iraq.

Table 6 describes the responses of university academics to different questions about their research practice. A significant difference was found between the different holders of academic titles and the time allocated for conducting research. Professors devoted the highest percentage of time (50%) to research, while the other three academic title holders devoted only 25% of their time to conducting research. Although there was a significant difference between the different academic title holders in their motivation to conduct research, they all considered

**Table 2. Satisfaction and motivation for conducting research.**

| Items | No. | (%) |
|---|---|---|
| **Research and publishing papers are a part of my job** | | |
| No | 51 | (5.4) |
| Yes | 898 | (94.6) |
| **Satisfaction with my academic role as a researcher** | | |
| No | 459 | (48.4) |
| Yes | 490 | (51.6) |
| **If no, reasons for dissatisfaction** | | |
| Having a safe Academic position without research | 15 | (1.6) |
| No Academic motivation | 99 | (10.4) |
| No financial motivation | 197 | (20.8) |
| No research cores | 22 | (2.3) |
| Poor infrastructure | 150 | (15.8) |
| Other | 85 | (9.0) |
| **The main motivation for conducting research** | | |
| Getting a Higher Diploma | 35 | (3.7) |
| Getting Academic Promotion | 258 | (27.2) |
| Job responsibility | 145 | (15.3) |
| Passion for Science and Continues Learning | 511 | (53.8) |

passion for science the main motivation. There is a significant difference between different holders of academic titles regarding satisfaction with their academic role as researchers and the motivations for conducting research. Only one professor has not published in a national

**Table 3. Publications and grant applications.**

| Items | No. | (%) |
|---|---|---|
| **Number of published research papers inside the country (Iraqi journals)** | | |
| 1–5 | 504 | (53.1) |
| 6–10 | 149 | (15.7) |
| 11–15 | 49 | (5.2) |
| More than 15 | 47 | (5.0) |
| None | 200 | (21.1) |
| **Number of published research papers outside the country (International journals)** | | |
| 1–5 | 453 | (47.7) |
| 6–10 | 102 | (10.7) |
| 11–15 | 41 | (4.3) |
| More than 15 | 66 | (7.0) |
| None | 287 | (30.2) |
| **Participation in any workshop to write a research proposal to support a grant application** | | |
| No | 472 | (49.7) |
| Yes | 477 | (50.3) |
| **Applying for any international grants for research** | | |
| No | 817 | (86.1) |
| Yes | 132 | (13.9) |
| **Research collaborations, either locally or internationally** | | |
| No | 300 | (31.6) |
| Previously I had | 140 | (14.8) |
| Yes | 509 | (53.6) |

**Table 4. The attitude of the role of university academics of the Ministry of Higher Education and Scientific Research in improving research and the use of research results by stakeholders.**

| Items | No. | (%) |
|---|---|---|
| **Ministry of Higher Education and Scientific Research has a sufficient vision, mission, and plan to conduct effective research by academic staff** | | |
| No | 454 | (47.8) |
| Somehow | 374 | (39.4) |
| Yes | 121 | (12.8) |
| **Research results were used by the related local authorities to improve or solve any problems in Kurdistan Region of Iraq** | | |
| No | 507 | (53.2) |
| Yes | 446 | (46.8) |
| **Shared research findings with relevant ministries or institutions by academics** | | |
| No | 504 | (53.1) |
| Yes | 448 | (46.9) |
| **Ways of sharing research findings* (n = 448)** | | |
| Sharing the paper or journal | 266 | (59.4) |
| Sharing a summary in the Kurdish language | 60 | (13.4) |
| Seminar | 189 | (42.2) |
| Workshop | 85 | (19.0) |
| Conference | 159 | (35.5) |
| Project | 85 | (19.0) |
| Other | 65 | (14.5) |

*total = more than one option

journal, whereas around half of assistant lecturers have not published any articles in national or international journals.

Although there is a statistically significant difference between different holders of academic titles, all believe that the Ministry of Higher Education and Scientific Research lacks a clear vision, mission, and plan to conduct effective research. There was a significant difference between the authors of academic titles in sharing research findings and collaboration (Table 7).

## Discussion

The present study examined the situation of research practice among university academics in Kurdistan Region of Iraq universities concerning practice, satisfaction, motivation, and

**Table 5. Challenges in conducting effective and productive research.**

| Items | No. | (%) |
|---|---|---|
| **Main challenges facing conducting effective research** | | |
| Lack of Funding | 596 | (62.8) |
| Lack of research infrastructure and technology | 453 | (47.7) |
| Lack of industry-academia cooperation | 335 | (35.3) |
| No support and encouragement | 489 | (51.5) |
| Other | 145 | (15.3) |
| **Research carried out by Kurdistan academic scholars is effective and produces new knowledge that is recognized worldwide** | | |
| No | 332 | (35.0) |
| Yes | 617 | (65.0) |

**Table 6. Comparison between different holders of academic titles with their research practice and the level of satisfaction.**

| Items | Assistant lecturer | Lecturer | Assistant Professor | Professor | P value |
|---|---|---|---|---|---|
| | No. (%) | No. (%) | No. (%) | No. (%) | |
| **The proportion of time assigned to research** | | | | | |
| Less than 25% | 97(29.5) | 90(26.4) | 40(17.8) | 1(1.9) | <0.001 |
| 25% | 119(36.2) | 109(32.0) | 82(36.4) | 10(18.5) | |
| 50% | 72(21.9) | 95(27.9) | 67(29.8) | 33(60.1) | |
| 75% | 33(10.0) | 44(12.9) | 31(13.8) | 6(11.1) | |
| 100% | 8(2.4) | 3(0.9) | 5(2.2) | 4(7.4) | |
| **Conducting and publishing research is part of the academic job** | | | | | |
| Yes | 304(92.4) | 328(96.2) | 212(94.2) | 54(100) | 0.046 |
| No | 25(7.6) | 13(3.8) | 13(3.8) | 0(0.0) | |
| **Satisfaction with an academic role as a researcher** | | | | | |
| Yes | 165(50.2) | 170(49.9) | 116(51.6) | 39(72.2) | 0.020 |
| No | 164(49.8) | 171(50.1) | 109(48.4) | 15(27.8) | |
| **The main motivation for conducting research** | | | | | |
| Obtain a higher diploma | 20(6.1) | 9(2.6) | 6(2.7) | 0(0.0) | <0.001 |
| Academic promotion | 110(33.4) | 106(31.1) | 42(18.7) | 0(0.0) | |
| Job responsibility | 45(13.7) | 53(15.5) | 36(16.0) | 11(20.4) | |
| Passion for science | 154(46.8) | 173(50.7) | 141(62.7) | 43(79.6) | |
| **Participation in workshops and training for grant proposal writing** | | | | | |
| Yes | 163(49.5) | 166(48.7) | 118(52.4) | 30(55.6) | 0.695 |
| No | 166(50.5) | 175(51.3) | 107(47.6) | 24(44.4) | |
| **Application for international grant** | | | | | |
| Yes | 20(6.1) | 57(16.7) | 47(20.9) | 8(14.8) | <0.001 |
| No | 309(93.9) | 284(83.3) | 178(79.1) | 46(85.2) | |
| **Number of published papers in national journal** | | | | | |
| None | 137(41.6) | 54(15.8) | 8(3.6) | 1(1.9) | <0.001 |
| 1–5 | 187(56.8) | 229(67.2) | 78(34.7) | 10(18.5) | |
| 6–10 | 3(0.9) | 49(14.4) | 89(39.6) | 8(14.8) | |
| 11–15 | 0(0.0) | 6(1.8) | 35(15.6) | 8(14.8) | |
| More than 15 | 2(0.6) | 3(0.9) | 15(6.7) | 27(50) | |
| **Number of published papers in international journals** | | | | | |
| None | 173(52.6) | 88(25.8) | 26(11.6) | 0(0.0) | <0.001 |
| 1–5 | 144(43.8) | 175(51.3) | 118(52.4) | 16(29.6) | |
| 6–10 | 8(2.4) | 44(12.9) | 42(18.7) | 81(14.8) | |
| 11–15 | 2(0.6) | 19(5.6) | 13(5.8) | 7(13) | |
| More than 15 | 2(0.6) | 15(4.4) | 26(11.6) | 23(42.6) | |

challenges. Poor access to publications was an important obstacle to publishing research, with only 8% of the respondents having access through institutional subscriptions. There is research evidence for a positive effect of online access to publications on publication output in universities in developing countries [20]. Unfortunately, most universities in Kurdistan Region of Iraq do not have subscriptions to the main publishers' databases [21]. Even the universities with subscriptions are limited to some freely available or subsidized databases for developing countries that do not provide full coverage to the main publishers [22]. Moreover, many universities lack adequate IT facilities and infrastructure to provide enough computers and the Internet to help them access online databases. Universities in developing countries have rarely been able to subscribe to academic journals in the past [20]. In recent years, several initiatives have been

**Table 7. Comparison between different academic titleholders with their attitude regarding research support and usage.**

| Items | Assistant lecturer | Lecturer | Assistant Professor | Professor | P value |
|---|---|---|---|---|---|
| | No. (%) | No. (%) | No. (%) | No. (%) | |
| **The Ministry of Higher Education and Scientific Research has a clear vision, mission, and plan for conducting effective research** | | | | | |
| Yes | 44(13.4) | 49(14.4) | 19(8.4) | 9(16.7) | 0.003 |
| Somehow | 144(43.8) | 125(36.7) | 77(34.2) | 28(51.9) | |
| No | 141(42.6) | 167(49.0) | 129(57.3) | 17(31.5) | |
| **Usage of research outcomes by local authorities** | | | | | |
| Yes | 154(46.8) | 164(48.1) | 105(46.5) | 22(40.7) | 0.796 |
| No | 175(53.2) | 177(51.9) | 120(53.3) | 32(59.3) | |
| **Sharing research findings with stakeholders** | | | | | |
| Yes | 116(35.3) | 184(53.8) | 118(52.4) | 31(57.4) | 0.011 |
| No | 213(64.7) | 157(46.2) | 107(47.6) | 23(42.6) | |
| **Kurdistan research is effective and produces new knowledge** | | | | | |
| Yes | 211(64.1) | 114(33.3) | 141(62.7) | 38(70.4) | 0.637 |
| No | 118(36.9) | 227(66.7) | 84(37.3) | 16(29.6) | |
| **Research collaboration (local and/or international)** | | | | | |
| Yes | 143(43.5) | 202(59.4) | 126(56.0) | 38(70.4) | 0.001 |
| Previously I had | 42(12.8) | 48(14) | 43(19.1) | 7(13) | |
| No | 144(43.8) | 91(26.6) | 56(24.9) | 9(16.7) | |

implemented worldwide to facilitate researchers' access to scientific literature in developing countries [22]. Unfortunately, many universities and researchers in developing countries, including those in Kurdistan Region of Iraq, might not be aware of the presence of such initiatives.

The present study revealed that only half of the participants were satisfied despite accepting conducting research and publishing articles as part of their jobs. Lack of financial support and academic motivation, and poor infrastructure were the most frequent reasons for dissatisfaction. The university academics identified a passion for science and continuous learning as the most motivator. Lack of motivation could be related to the political instability in the region and personal factors such as lack of mentorship, isolation, and competing priorities. The presence of a large number of predatory journals and corruption in research and publications can also demotivate researchers. A study from the UK revealed that about 65% of university academics were satisfied, very satisfied, or extremely satisfied with the research, which is higher than the results of the present study [23]. Research conducted on 94 academic staff in South Africa revealed that less than 50% were generally satisfied with research questions. Factors that caused dissatisfaction included government interference in teaching, poor quality of students' work; research-related aspects such as lack of time to do research, shortage of research assistants, uncertainty about how to do research and the quality of their research efforts; promotion criteria and politics surrounding promotion; time spent on administrative work, the amount of paperwork involved, and the level of interaction at meetings; poor academic communication among colleagues; salaries compared to salaries outside the higher education system, lack of funding to attend conferences, and lack of recognition for work within the institution [24]. A study from Poland revealed that the level of satisfaction of researchers with their scientific work depends on the employment conditions and the social importance of the research carried out. The level of satisfaction from work is closely correlated with the scientific opportunities of researchers (that is, the possibility of academic and didactic work and contact with students and coworkers) and negatively correlated with the necessity to carry out administrative work.

Most Polish researchers were proud of their scientific achievements and treated their profession as a passion or vocation [25].

Exploring research and teaching self-efficacy and job satisfaction of 528 university faculty (46% female) from Azerbaijan and Turkey using a mixed methods approach indicated that teaching self-efficacy was higher than research self-efficacy and that research self-efficacy varied according to career stage and qualifications, but not gender. Job satisfaction was highest for faculty members with Master's degrees. Teaching self-efficacy was the strongest predictor of job satisfaction. Furthermore, contextual factors such as university climate and peer collegiality influenced self-efficacy and job satisfaction [26].

Lack of financial support was the most frequent reason for dissatisfaction among the researchers in the present study. Conducting and publishing research is costly and needs financial support. Unfortunately, there are limited funding opportunities for research in Kurdistan Region of Iraq, particularly from industry and commerce [27, 28]. The public and private universities in the region provide some financial support to the researchers in return for publishing papers that they encourage for university ranking purposes. However, such support is limited to publication fees and is not regular, particularly for the public universities, due to the current financial crises in the region. The current financial crisis and the dispute on budget between the Kurdistan Regional Government and the Iraqi Central Government have led to significant delays in the payment of the monthly salaries of the governmental employees [29]. Thus, researchers from public universities can not afford to pay personally to conduct and publish research.

The present study revealed that around half of the academic staff had not published more than five research articles in journals within or outside the country. Very few academic staff applied for international grants, although around half had local or international research collaboration. The most critical challenges for 53 government and university executives and academics in the Philippines were effectively meeting the dual demands of teaching and research, building a critical mass of researchers, and developing excellent research skills and competencies among staff and students [30]. A qualitative research method was conducted on 34 academics working at a university in Turkey. According to the results obtained from the research, it was determined that academics had foreign language problems, difficulties in the data analysis process, problems in publishing their research, time problems in their research, problems in collaborative work with their colleagues, and difficulties in reaching international resources [31].

In the present study, getting a scientific promotion was only a minor motivator for the university academics to conduct research. The current regulations for scientific promotion are of the old style (from 2016) that limit the number of publications needed for each level of promotion. They do not account for having more publications beyond the specified numbers [28]. There is also a lack of specific incentives or requirements to publish research after the highest rank of professorship, with a lack of professor emeritus title in the regulations. Moreover, no explicit rules exist against the academic staff with no publications or scientific promotion, except for a clause for assistant lecturers, which is not enforced [27].

Almost half of the study sample in the present study believed that the Ministry of Higher Education and Scientific Research has no sufficient vision, mission, or plan for conducting effective research by academic staff. Similarly, another study from Kurdistan Region of Iraq revealed a lack of a clear strategy from the government to engage researchers, academic staff, and universities in organized research plans [28]. The lack of a national research council in the region is an important factor in the lack of such a clear research strategy. An important challenge the researchers face in Kurdistan Region of Iraq is the frequent changes in the rules and regulations by the Ministry of Higher Education and Scientific Research. While research and

publications are encouraged for different purposes, such as scientific promotion, university ranking, and postgraduate studies requirements, the frequently amended rules and requirements affect the quality of higher education and postgraduate studies in Kurdistan Region of Iraq.

More than half of the study sample in the present study mentioned that related governmental authorities do not use their research results. Palamarchuk (2018), regarding the functions of academic staff in the effective governance of universities, mentioned that in addition to the basic knowledge of the system, governance structure, governance principles, core values, and qualities of academic staff necessary to perform their functions and responsibilities in the governing bodies, academic staff also perform their immediate and top priority functions in the university, which is the function of learning, teaching, and research [32]. There is a lack of close collaboration between the Ministry of Higher Education and Scientific Research and the other relevant ministries, particularly the Ministry of Planning, for research products and using research findings in policymaking [33]. There is also no effective communication among researchers, the Ministry of Higher Education and Scientific Research, and the other ministries. These factors discourage the researchers from producing and publishing research as they believe policymakers do not listen to their voices.

A qualitative study in Indonesia to explore the role of college leadership in cultivating research showed that the role of the university leader in research study can be played by providing funding support, spreading research information, motivating, retrieving appropriate policies related to the research activities of lecturers, establishing cooperation, facilitating civitas in research, and taking a humanistic approach [34].

Although no study indicates the role of academic research in developing the quality of governance of Kurdistan Region of Iraq and the relationship of higher education with other ministries, a considerable gap is felt. This may be due to the unstable economic and political situation in the region, which is related to several wars and conflicts, and the low experience of the Kurdistan Government (1991). A study from Indonesia indicated that universities, industries, and the government operate as three independent spheres, still quite distant from each other. The researchers found only a small number of examples in which the three spheres cooperated productively and in which universities developed and shared essential knowledge with the other spheres. Only one case in which actors from the three sectors developed a new organizational structure to work together to generate and implement joint ideas and strategies. However, the initiative was an isolated example and not a regular feature of the regional innovation system. The results also showed that none of the three spheres is sufficiently equipped to lead the development of Indonesia's innovation systems. International study visits to Korea and China showed that cooperative initiatives are being driven by their respective governments [35].

The present study has several limitations inherited from this study design of online questionnaire surveys. The inability to select a proper random sample of the study population and possible contamination of the sample by having self-selected respondents with biases in the sample might affect the generalizability of the findings. We could not define causality because this was a cross-sectional study. The information from respondents was obtained by self-reported questionnaire, which might result in biases or inaccuracies in the data. A longitudinal study that applies random sampling and considers confounding variables is needed to ensure the generalizability of the findings. With the lack of a previously validated questionnaire for this type of study, a new questionnaire was developed and used. Although this new questionnaire was validated, it will not work as previously validated and used questionnaire in preventing measurement errors.

## Conclusions

The university academics of Kurdistan Region of Iraq are passionate about their role as researchers but have many challenges in conducting effective research. Lack of access to subscribed databases, not using the research findings by the relevant authorities, and lack of clear governmental strategy to support research are important challenges facing universities and researchers that hinder research production. The Ministry of Higher Education and Scientific Research needs to have a strategic plan to improve and foster an environment for academic staff in terms of infrastructure, finance, and technical, as well as clarify the vision and mission of the ministry. The ministry must also update the scientific promotion regulations to include other important aspects to the limited number of publications, such as obtaining research grants and funding and international research cooperation. There is also a need to establish a national research center to support and direct researchers and bridge the gaps between researchers and policymakers in the region. More studies are needed to identify the root factors of the needs and challenges of academic staff.

## Supporting information

**S1 Appendix. Study questionnaire.**
(DOCX)

## Author Contributions

**Conceptualization:** Hamdia Mirkhan Ahmed, Nazar P. Shabila, Amir Abdulrahman Ahmad.

**Data curation:** Hamdia Mirkhan Ahmed, Nazdar Ezzaddin Alkhateeb, Amir Abdulrahman Ahmad.

**Formal analysis:** Nazdar Ezzaddin Alkhateeb.

**Methodology:** Hamdia Mirkhan Ahmed, Nazdar Ezzaddin Alkhateeb, Nazar P. Shabila.

**Supervision:** Amir Abdulrahman Ahmad.

**Writing – original draft:** Hamdia Mirkhan Ahmed, Nazdar Ezzaddin Alkhateeb, Nazar P. Shabila, Amir Abdulrahman Ahmad.

**Writing – review & editing:** Nazar P. Shabila.

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
