## [Decision Letter · Decision Letter 0]

4 Apr 2024

PONE-D-24-10517Research practice, satisfaction, motivation, and challenges among university academics in Kurdistan Region of IraqPLOS ONE

Dear Dr.Nazar P. Shabila

Thank you for submitting your manuscript to PLOS ONE. After careful consideration, we feel that it has merit but does not fully meet PLOS ONE’s publication criteria as it currently stands. Therefore, we invite you to submit a revised version of the manuscript that addresses the points raised during the review process.

We look forward to receiving your revised manuscript.

Kind regards,

Ranjdar Mahmood Talabani

Academic Editor

PLOS ONE

Journal Requirements:

Reviewers' comments:

Reviewer's Responses to Questions

**Comments to the Author**

1. Is the manuscript technically sound, and do the data support the conclusions?

Reviewer #1: Yes

2. Has the statistical analysis been performed appropriately and rigorously? 

Reviewer #1: Yes

3. Have the authors made all data underlying the findings in their manuscript fully available?

Reviewer #1: No

4. Is the manuscript presented in an intelligible fashion and written in standard English?

Reviewer #1: Yes

5. Review Comments to the Author

Reviewer #1: Review for Manuscript ID:

PONE-D-24-10517 entitled "Research practice, satisfaction, motivation, and challenges among university academics in Kurdistan Region of Iraq"

In general, the paper is well-written and easy to follow. The authors highlight an issue that is of real concern. Thus, I admire their initiative. There are some minor points that need to be addressed, as follows.

General notes:

1. I believe one of the factors that affects research productivity is access to scientific journals and books. In the Kurdistan region, the majority of universities, public ones in particular, do not provide electronic access to their academic staff. This is something that, I believe, has to be discussed in the manuscript.

2. There is financial support for publication by some universities (some public and the majority of private universities). Again, this has to be explored.

3. The authors have to mention the rules by the Ministry of Higher Education in KRG as these rules are amended frequently and affect the quality of higher education in KRG.

4. There needs to be a clearer collaboration between the Ministry of Higher Education and the Ministry of Planning in KRG. This discourages academic staff from doing research, as they believe that policymakers in the KRG do not hear their voices. Thus, it has to be added to the discussion section.

5. The lack of a clear rule for academic staff with no publication and an old style of academic title promotion (the latest rule is from 2016 in KRG) could also be related to the fact that research publication is not taken seriously by the academic staff. Again, this has to be explored.

6. The salary issue should also be mentioned, as government employees in KRG suffer delays in their monthly salary KRG government.

Specific notes:

1. A copy of the questionnaire has to be added as a figure in the method section.

2. Is the questionnaire validated? This has to be stated in the methodology.

3. The ethical committee's number and approval date must be added.

4. The methods sections should be sectioned as follows: study design, questionnaire, participants, and statistical analysis.

5. What are the study's limitations? This has to be added to the discussion. There are inherent issues with the questionnaire study.

6. Table 1 better to be presented as a pie chart.

BW,

6. PLOS authors have the option to publish the peer review history of their article (what does this mean?). If published, this will include your full peer review and any attached files.

Reviewer #1: **Yes: **Sarhang Sarwat Gul

---

## [Author Response · Author response to Decision Letter 0]

6 Apr 2024

Reviewer 1

In general, the paper is well-written and easy to follow. The authors highlight an issue that is of real concern. Thus, I admire their initiative. 

There are some minor points that need to be addressed, as follows.

Authors’ response

Thank you very much for the positive comments. Thank you for the comments and suggestions that we found very useful

General notes:

1. I believe one of the factors that affects research productivity is access to scientific journals and books. In the Kurdistan region, the majority of universities, public ones in particular, do not provide electronic access to their academic staff. This is something that, I believe, has to be discussed in the manuscript.

Authors’ response

Thank you very much for this valuable comment. The issue of access to scientific journals and books is now thoroughly discussed in the Discussion section (Pages 12-13, lines 211-224). It is also included in the Conclusions (Page 18, lines 346-347).

2. There is financial support for publication by some universities (some public and the majority of private universities). Again, this has to be explored.

Authors’ response

The issue of financial support for research, the support by some universities, and related issues are now discussed in detail in the Discussion section (Page 14, lines 260-264).

3. The authors have to mention the rules by the Ministry of Higher Education in KRG as these rules are amended frequently and affect the quality of higher education in KRG.

Authors’ response

The ministry's main rules and the effect of their frequent amendments on higher education are now discussed in detail (Page 16, lines 291-299).

4. There needs to be a clearer collaboration between the Ministry of Higher Education and the Ministry of Planning in KRG. This discourages academic staff from doing research, as they believe that policymakers in the KRG do not hear their voices. Thus, it has to be added to the discussion section.

Authors’ response

Thanks for this comment. We have now discussed the issue of collaboration and knowledge transfer in detail in the Discussion section (Page 16, lines 307-312). It is also included in the Conclusions (Page 18, lines 347, 355-356).

5. The lack of a clear rule for academic staff with no publication and an old style of academic title promotion (the latest rule is from 2016 in KRG) could also be related to the fact that research publication is not taken seriously by the academic staff. Again, this has to be explored.

Authors’ response

This lack of clear rules for academic staff with no publication is now discussed in detail in the Discussion section (Page 15, lines 281-288). It is also included in the Conclusion (Page 18, lines 352-354).

6. The salary issue should also be mentioned, as government employees in KRG suffer delays in their monthly salary KRG government.

Authors’ response

The issue of delays in staff salary is now discussed in the Discussion section (Page 14-15, lines 264-268).

Specific notes:

1. A copy of the questionnaire has to be added as a figure in the method section.

Authors’ response

A copy of the questionnaire is now provided as the supporting information file (Page 5, lines 118-119 and S1 Appendix file).

2. Is the questionnaire validated? This has to be stated in the methodology.

Authors’ response

Details are provided about the validation of the questionnaire (Page 6, lines 120-123).

3. The ethical committee's number and approval date must be added.

Authors’ response

The ethical committee approval number and date are provided (Page 6, line 137).

4. The methods sections should be sectioned as follows: study design, questionnaire, participants, and statistical analysis.

Authors’ response

The Methods section is now divided to sub-sections, as advised (Page 5-6, lines 109, 112, 127, and 131).

5. What are the study's limitations? This has to be added to the discussion. There are inherent issues with the questionnaire study.

Authors’ response

The study limitations are now provided and discussed in detail at the end of the Discussion section (Pages 17-18, lines 333-343).

6. Table 1 better to be presented as a pie chart.

Authors’ response

Table 1 is changed to a pie chart (Page 7, line 145 and Figure 1).

---

## [Editor Report · Decision Letter 1]

9 Apr 2024

Research practice, satisfaction, motivation, and challenges among university academics in Kurdistan Region of Iraq

PONE-D-24-10517R1

Dear Dr. Nazar P. Shabila

We’re pleased to inform you that your manuscript has been judged scientifically suitable for publication and will be formally accepted for publication once it meets all outstanding technical requirements.

Kind regards,

Ranjdar Mahmood Talabani

Academic Editor

PLOS ONE